# Development and validation of an assay for detection of Japanese encephalitis virus specific antibody responses

**Pradeep Darshana Pushpakumara[1], Chandima Jeewandara[2], Laksiri Gomes[2], Yashodha Perera[2], Ananda Wijewickrama[3], Gathsaurie Neelika Malavige[2‡], Charitha Goonesekara[1‡]***

**1** General Sir John Kotelawala Defence University, Colombo, Sri Lanka, **2** Centre for Dengue Research, University of Sri Jayewardenepura, Nugegoda, Sri Lanka, **3** National Institute of Infectious Diseases, Angoda, Sri Lanka

‡ These authors are joint senior authors on this work.
* charithalg@kdu.ac.lk

## Abstract

### Introduction

Although immune responses to the Japanese Encephalitis virus (JEV), and the dengue viruses (DENV) have a potential to modulate the immune responses to each other, this has been poorly investigated. Therefore, we developed an ELISA to identify JEV specific, DENV non cross-reactive antibody responses by identifying JEV specific, highly conserved regions of the virus and proceeded to investigate if the presence of JEV specific antibodies associate with dengue disease severity.

### Methodology and results

22 JEV specific peptides were identified from highly conserved regions of the virus and the immunogenicity and specificity of these peptides were assessed in individuals who were non-immune to JEV and DENV (JEV⁻DENV⁻, N = 30), those who were only immune to the JEV and not DENV (JEV⁺DENV⁻, N = 30), those who were only immune to DENV(JEV⁻⁻ DENV⁺, N = 30) and in those who were immune to both viruses (JEV⁺DENV⁺, N = 30). 7/22 peptides were found to be highly immunogenic and specific and these 7 peptides were used as a pool to further evaluate JEV-specific responses. All 30/30 JEV⁺DENV⁻ and 30/30 JEV⁺⁻ DENV⁺ individuals, and only 3/30 (10%) JEV⁻DENV⁺ individuals responded to this pool. We further evaluated this pool of 7 peptides in patients following primary and secondary dengue infection during the convalescent period and found that the JEV-specific peptides, were unlikely to cross react with DENV IgG antibodies. We further compared this in-house ELISA developed with the peptide pool with an existing commercial JEV IgG assay to identify JEV-specific IgG following vaccination, and our in-house ELISA was found to be more sensitive. We then proceeded to investigate if the presence of JEV-specific antibodies were associated with dengue disease severity, and we found that those who had past severe dengue (n

**Data Availability Statement:** All relevant data are within the manuscript and its Supporting Information files.

**Funding:** Funding was provided by the Centre for Dengue Research and National Science Foundation, Sri Lanka (RG/2015/HS/07).

**Competing interests:** The authors have declared that no competing interests exist.

= 175) were significantly more likely (p<0.0001) to have JEV-specific antibodies than those with past non-severe dengue (n = 175) (OR 5.3, 95% CI 3.3 to 8.3).

## Conclusions

As our data show that this assay is highly sensitive and specific for detection of JEV-specific antibody responses, it would be an important tool to determine how JEV seropositivity modulate dengue immunity and disease severity when undertaking dengue vaccine trials.

## Introduction

Infections due to the flaviviruses such as the dengue viruses (DENV) and the Japanese encephalitis virus (JEV), are a major cause of morbidity and mortality worldwide. The DENV infects 390 million individuals annually resulting in over 25,000 deaths [1]. Epidemics due to JEV are uncommon due to the availability of a safe and effective vaccine for its prevention [2]. Although, there is a live attenuated vaccine licensed for the prevention of dengue, its efficacy varies for different DENV serotypes and the manufacturer has cautioned that it may cause more severe disease in dengue naïve individuals [3]. However, due to the cross-reactive nature of antibodies generated to the JEV and DENV, there is a possibility of modulation of dengue-specific immune responses by immune responses to JEV [4]. Therefore, in order to develop safer and more efficacious vaccines, it is important to have a better understanding of how immunity to JEV could influence the immunity to the DENV.

Primary infections with a particular DENV serotype usually results in mild or asymptomatic disease, while a subsequent infection (secondary dengue infection) with other DENV serotype may lead to severe forms of disease, such as dengue hemorrhagic fever (DHF) [5]. This is thought to be due to phenomenon known as antibody dependent enhancement (ADE), where sub-neutralising concentrations of antibodies produced against previous DENV serotype cross-react with the infecting serotype, thereby enhancing DENV infection [6, 7]. Both DENV and JEV are genetically and antigenically closely related and share 54.3% amino acid homology within the envelope protein [8]. Therefore, due to the similarities between the two viruses, antibodies generated towards JEV may cross react with the DENV and lead to disease enhancement [9, 10].

Several previous studies have suggested that pre-existing antibodies to JEV were associated with increased occurrence of symptomatic DENV infection [9–11]. In contrast, studies in mouse models have shown that mice immunized with the JE vaccine were subsequently protected against dengue and also that individuals who had received the JEV vaccine had antibodies that cross-neutralized certain DENV serotypes [12]. Previously, we found that those who had been hospitalized due to dengue were more likely to be seropositive for JEV, when compared to those who had mild/asymptomatic dengue [10]. However, the JEV-specific IgG ELISA used in this study was not specific to JEV, and the increased JEV seropositivity, could be a mere reflection of the presence of JEV cross-reactive DENV antibodies [10]. The current commercial assays cannot differentiate JEV-specific antibodies from those due to the DENV cross-reactivity of antibodies [13, 14] and the plaque reduction neutralization assays, which are the gold-standard for detection of JEV-specific antibodies are cumbersome to perform [15]. Therefore, in order to determine if the presence of JEV-specific antibodies influences the outcome of subsequent dengue infections, it would be important to use antibody assays that can differentiate JEV-specific antibody responses from those due to DENV.

In Sri Lanka, immunization against the JEV started in a phase by phase basis, and since 1988 was subsequently included in the national routine immunization program [16]. The Nakayama strain of the inactivated JE vaccine was used from 1988 to1992, the Beijing strain of the inactivated JE vaccine from 1992 to 2006 and the live attenuated vaccine (LJEV SA14-14-2) given since 2006. Therefore, it would be also important to determine if pre-existing immunity to the JEV through vaccination, led to any difference in dengue disease severity subsequently in these individuals.

In this study, we identified a panel of peptides which are highly conserved within the JEV genome and did not significantly cross-react with the DENV. Using these peptides, we developed an ELISA to detect JEV-specific antibodies and proceeded to investigate if those with JEV-specific antibodies were more likely develop severe dengue infection.

## Materials and methods

### Identification JEV-specific of highly conserved regions within JEV and development of a JEV-IgG ELISA

One hundred and seventeen JEV polyprotein sequences, which were isolated within a period of 50 years from the South Asian and South East Asian region were retrieved from National Center for Biotechnology information. These sequences were aligned using ClustalW, on Mega 7 software (www.megasoftware.net/) to identify the degree of conservation. Regions which showed >90% conservation were identified and sectioned into 20mers peptides (36 peptides) overlapping by 5 or 10 amino acids. The specificity of these JEV peptides was determined by using Clustal Omega of European Bioinformatics Institute (EBI) (www.ebi.ac.uk) to confirm that they did not significantly cross-react with the DENV (S1 Table). Out of these 36 peptides, only 22 peptides were successful in the synthesis with 90% purity (Genscript USA) and were used for further analysis as previously described [17].

### Recruitment of healthy individuals for identification of JEV-specific antibodies

As a part of our previous study, we had recruited 1689 individuals, between the ages of 6 to 80 years, who were registered at the Family Practice Centre of the University of Sri Jayewardene-pura for provision of their primary health care [10]. We studied IgG antibody responses to the 22 JEV-specific peptides in 120/1689 healthy individuals who were randomly selected from the initial cohort based on their immunity to JEV and DENV. Information such as if they received the JEV vaccine was recorded at the time of recruitment. Of these 120 individuals, those who had not received the JEV vaccine and were also seronegative for JEV IgG antibodies by a commercial ELISA (Inbios, ISA), were considered as JEV seronegative (JEV⁻). Those who had received the JEV vaccine and who had detectable JEV IgG antibodies by a commercial ELISA were considered as JEV seropositive (JEV⁺). Individuals who had received the JEV vaccine and were seronegative by the commercial JEV IgG ELISA or those who had not received the JEV vaccine and were seropositive based on the commercial JEV IgG ELISA were not considered in the analysis. The responses to these 22 peptides were assessed in JEV vaccinated, DENV seronegative individuals (JEV⁺DENV⁻, n = 30), DENV seropositive but JEV nonvaccinated individuals (JEV⁻DENV⁺, n = 30), JEV vaccinated and DENV seropositive individuals (JEV⁺⁻DENV⁺, n = 30) and individuals seronegative for DENV and nonvaccinated for JEV (JEV⁻⁻DENV⁻, n = 30). To find out if the JEV seropositivity was associated with severe dengue (SD), we evaluated JEV specific responses in 175 individuals who had developed severe dengue in the past (SD) and 175 individuals who were seropositive for dengue but never showed any

symptoms and were thus considered to have a past inapparent dengue or only had dengue fever in the past (NSD), within this group of 1689 individuals. The detailed methods of recruitment and follow up are described in S1 File.

In this total cohort of individuals of 520 individuals (the 120 individuals whose sera was used to study JEV-specific immune responses, the sera of 50 individuals which was used to study the JEV specific peptides and 175 individuals with past SD and 175 individuals with past NSD), we further evaluated the age stratified seroprevalence of JEV-specific antibodies using the pool of JEV-specific peptides defined by us and also compared the JEV-IgG antibodies as detected by the Inbios ELISA.

## Recruitment of patients with acute dengue

69 adult patients with varying severity of acute dengue infection were recruited from the National Institute of Infectious Disease, Sri Lanka following informed written consent. The day on which the patient first developed fever was considered day one of illness. Blood samples were obtained during the febrile phase and again during the convalescent phase (day 21–30 during infection). All clinical and laboratory features were recorded serially. Clinical disease severity was classified according to the 2011 World Health Organization (WHO) dengue diagnostic criteria [5]. Patients with a rise in haematocrit > 20% of baseline or clinical or ultrasound scan evidence of plasma leakage were classified as having DHF. Shock was defined as having cold clammy skin, along with a narrowing of pulse pressure of 20 mmHg. As such, 28 patients were classified as DHF and 41 patients were classified as DF.

## Ethics statement

Blood samples were obtained following informed written consent. Ethics approval was obtained from the Ethics Review Committee of University of Sri Jayewardenepura and the General Sir John Kotelawala Defence University.

## Determining DENV and JEV serostatus in healthy individuals

The seropositivity of individuals to the DENV had been assessed in these individuals using the indirect dengue IgG capture ELISA (Panbio, Australia) [10]. JEV direct IgG ELISA (InBios International, USA) was used to detect IgG antibodies to the JEV in this cohort [10]. Immune status to the JEV was calculated using the immune status ratio (ISR) according to the manufacturers' instructions. An ISR of >5 was considered positive; an ISR of 2–5 equivocal and an ISR of <2 was considered negative.

## Serotyping of DENV in patients with acute dengue

Acute dengue infection was confirmed by quantitative real time PCR and DENV viruses were serotyped and titres quantified by quantitative real time PCR as previously described [18]. Multiplex quantitative real-time PCR was performed as using the CDC real time PCR assay for detection of the dengue viruses [19], and modified to quantify DENV. Oligonucleotide primers and a dual labeled probe for DENV 1,2,3,4 serotypes were used (Life technologies, India) based on published sequences [19].

## Determining DENV serostatus in patients with acute dengue

Dengue antibody assays were performed using a commercial capture-IgM and IgG ELISA (Panbio, Brisbane, Australia) [20, 21]. Based on the WHO criteria, patients with an IgM: IgG ratio of >1.2 were considered to have a primary dengue infection, while patients with IgM:

IgG ratios <1.2 were categorized under secondary dengue infection [22]. The DENV-specific IgM and IgG ELISA was also used to semi-quantitatively determine the DENV-specific IgM and IgG titres, which were expressed as Panbio units.

## T cell-based assay to determine immunity to DENV serotypes

The past infecting serotype in dengue seropositive individuals in the community was carried out by using a T cell-based assay and previously described using a panel of DENV serotype specific peptides [23–25] (S1 File). Based on this assay those who were only immune to one DENV serotype were considered to have had a primary dengue infection with that particular serotype in the past. JEV-specific IgG antibody responses were measured in 10 individuals who were only immune to DENV1, 10 who were only immune to DENV2, 10 only immune to DENV3 and 10 only immune to DENV4.

## Development of an indirect ELISA to detect JEV-specific antibodies

Ninety-six-well microtitre plates (Pierce™ Cat: 15031) were coated with 100μl/well peptide preparations diluted in bicarbonate/carbonate coating buffer (pH 9.6) at a final concentration of 1μg/100μl. The peptides were coated individually and incubated overnight at 4 ˚C. After incubation, unbound peptide was washed away twice with phosphate-buffered saline (PBS, pH 7.4). The wells were blocked with 250μl/well PBS containing 0.05% Tween 20 and 1% bovine serum albumin (BSA) and incubated for 1.5 h at room temperature. The plates were then washed three times with 300μl/well washing buffer (PBS containing 0.05% Tween 20). Serum samples were diluted 1:250 in ELISA diluent (Mabtech, Sweden, Cat: 3652-D2). Diluted serum samples were added 100μl/well in duplicates and incubated for 30 min at room temperature. After incubation, wells were washed three times with 300μl/well washing buffer. Goat anti-human IgG, biotinylated antibody (Mabtech, Sweden Cat: 3820-4-250) was diluted 1:1000 in PBS containing 1% BSA and added 100μl/well. Plates were then incubated for 30 min at room temperature and washed three times. Streptavidin–HRP (Mabtech, Sweden, Cat: 3310–9) was diluted 1:1000 in PBS containing 1% BSA and added 100μl/well. Incubation was carried out for 30 min at room temperature, and the washing procedure was repeated five times. TMB ELISA substrate solution (Mabtech, Sweden, Cat: 3652-F10) was added 100μl/well and incubated in the dark for 10 min at room temperature. The reaction was stopped by adding 100μl/well 2 M $H_2SO_4$ stop solution and absorbance values were read at 450 nm.

## Statistical analysis

Statistical analysis was performed using GraphPad Prism version 6. As the data were not normally distributed, non-parametric tests were used in the statistical analysis and two-sided tests were carried out in all instances. Differences in mean values of IgG antibody responses to JEV-specific peptides in individuals with SD and NSD were compared using the Mann-Whitney U test (two tailed). Degree of association between past SD, NSD dengue infection and the presence of JEV peptide antibody responses was expressed as the odds ratio (OR), which was obtained from standard contingency table analysis by Haldane's modification of Woolf's method.

## Results

### Identification of JEV-specific peptides in those who were immune to JEV

Antibody responses to the 22 JEV-specific 20mer peptides in the JEV vaccinated, DENV sero-negative individuals (JEV[+]DENV[-], n = 30), DENV seropositive but JEV nonvaccinated

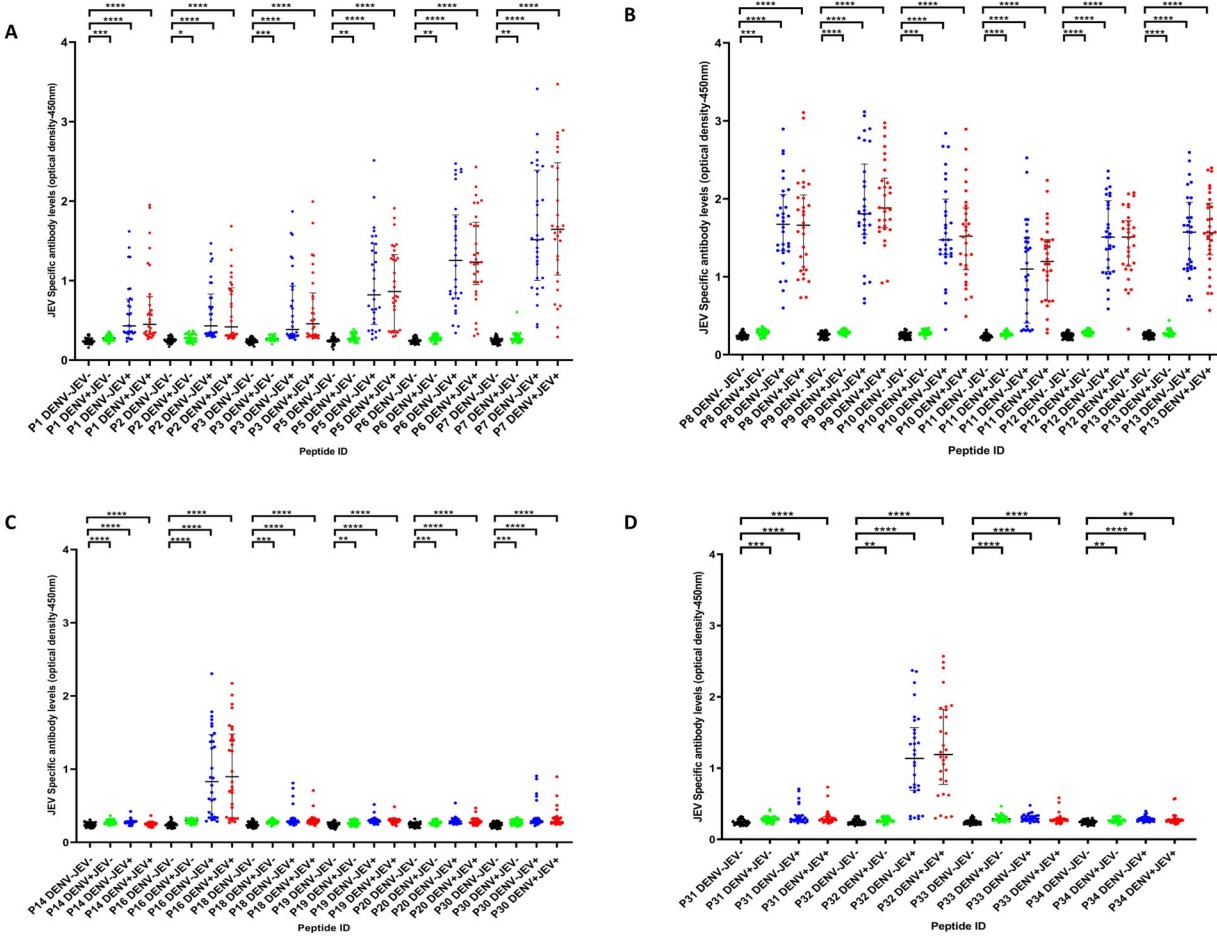

**Fig 1. IgG antibody levels to 22 JEV-specific peptides in individuals with varied DENV and JEV seropositivity.** JEV-specific antibody responses were measured by ELISA to 22 JEV-specific peptides in those who were both JEV and DENV seropositive (JEV+DEV+, n = 30), JEV seropositive but seronegative for DENV (JEV+DEV−, n = 30), DENV seropositive but JEV seronegative (JEV−DENV+, n = 30) or seronegative for both (JEV−DENV−, n = 30). Error bars indicate the median and the interquartile range. The horizontal dotted line represents the cut-off value of 0.382, which was considered as the mean, ±3SD of the optical density of antibody levels in JEV− and DENV− seronegative. *P<0.05, **P<0.01, ***P<0.001, ****P<0.0001. The differences in the responses to the JEV-specific antibodies were assessed by Mann Whitney two-tailed test.

individuals (JEV−DENV+, n = 30), JEV vaccinated and DENV seropositive individuals (JEV+−DENV+, n = 30) and individuals seronegative for DENV and nonvaccinated for JEV (JEV−−DENV−, n = 30) are shown in Fig 1. The cutoff value for a positive response for each peptide was calculated separately by calculating the mean±3SD of the optical density (OD) of antibody levels in DENV− and JEV− seronegative individuals (n = 30). Accordingly, the cut off value was calculated as 0.382 and all responses above this value were considered as a positive response.

Only one person out of the 30 JEV−DENV+ individuals responded to peptides 1, 5, 8, 13, and 30 and only 2/30 JEV−DENV+ individuals responded to peptide 2, 31, and 33. Responses to other JEV-specific peptides were not detected in any of the JEV−DENV+ individuals. The seven peptides represented by P6, P7, P8, P9, P10, P12 and P13 were the peptides that were predominantly recognized by the JEV+DENV− individuals (Fig 1). All (100%) of JEV+DENV− responded to P7, P8, P9, P12 and P13. 29/30 (96.7%) responded to P6 and 28/30 (93.3%) responded to P10 (Fig 1). All these seven peptides were located within the envelope protein. Out of these 7 peptides only P13 and P8 were recognized by a JEV−DENV+ individual, whereas P6, P7, P9, P10 and P12 peptides were not recognized by any of the JEV−DENV+ individuals.

P5, P11, P16 and P32 gave positive responses in 22/30 (73.3%) to 24/30 (80.0%) JEV$^+$DENV$^-$ individuals (S2 Table).

As the most immunodominant peptides were P6, P7, P8, P9, P10, P12, and P13 and were recognized by over 70% of individuals, we pooled these 7 peptides and then assessed the responses to this pool of peptides in JEV$^+$DENV$^-$ (n = 30), JEV$^-$DENV$^+$ (n = 30), JEV$^+$DENV$^+$ (n = 30) and JEV$^-$DENV$^-$ (n = 30) individuals. None of the JEV$^-$DENV$^-$ individuals responded to this JEV-specific peptide pool. All 30/30 (100%) JEV$^+$DENV$^-$ and JEV$^+$DENV$^+$ individuals, and only 3/30 (10%) JEV$^-$DENV$^+$ individuals responded to this pool of seven peptides (Fig 2A).

## Development of an in-house ELISA to detect JEV-specific IgG antibody responses

As P6, P7, P8, P9, P10, P12, and P13 were the most frequently recognized peptides by JEV immune, but DENV non-immune individuals (JEV$^+$DENV$^-$), we then sought to investigate the antibody responses to these peptides pooled together in the above 4 groups of individuals. We found that all individuals (100.0%) who were JEV vaccinated but DENV seronegative (JEV$^+$DENV$^-$) and all individuals (100.0%), who were JEV vaccinated seropositive for DENV (JEV$^+$DENV$^+$) responded to this pool of peptides (Fig 2A). None of JEV and DENV seronegative individuals (JEV$^-$DENV$^-$) responded to this peptide pool, whereas only 3/30 (10.0%) JEV$^-$DENV$^+$ individuals responded to this peptide pool (Fig 2A).

We then evaluated the suitability of this peptide pool to be used to differentiate JEV-specific IgG responses from those who were also seropositive for DENV. We considered the cut-off value for a positive result for this peptide pool as the mean ± 3 SD (standard deviations) of the OD of the sera of patients who were seronegative for both JEV and DENV. Accordingly, all responses to this peptide pool which were above 0.382 OD read at 450nm was considered as a positive result. All JEV$^+$ DENV$^-$ individuals had antibody levels >0.382 OD value which was considered as a positive cut-off value (Fig 2A).

According to the bioinformatics analysis these seven peptides have less than 36% homology with four DENVs (S1 Table and S1 Fig). Furthermore, according to the linear B cell epitope prediction tools (Ellipro-www.iedb.rg, Bepipred-www.iedb.org, BCEpredcrdd.osdd.net, and ABCpred- crdd.osdd.net; S3 Table), these seven peptides were found to be exposed to outside of the E protein (S2 Fig) and therefore, have higher potential to be immunogenic peptides.

## Cross reactivity of JEV-specific peptides in those with primary DENV infection

As these 7 peptides included in the peptide pool (P6, P7, P8, P9, P10, P12, and P13) had different homology with the four DENV serotypes, we proceeded to evaluate the possible cross reactivity of these peptides with those who had been infected with different DENV serotypes in the past. As it would be difficult to differentiate cross reactivity of this peptide pool in individuals who had been infected with multiple DENV serotypes in the past, we evaluated the cross reactivity of this JEV-specific peptide pool in those who had been infected with only one DENV serotype in the past. Using a T cell-based assay described above [23, 25], we evaluated the JEV peptide pool specific IgG responses, in individuals who had past infection with each DENV serotype. Only one person (1/10) who was JEV seronegative and was found to be immune to DENV1 serotype gave positive response, and 1/10 individuals who were seronegative for JEV but were immune to the DENV2 serotype gave a positive response to this peptide pool consisting of 7 peptides. In addition, 2/10 individuals who were seronegative for JEV but were immune to DENV3$^+$ also responded to this peptide pool, whereas none of the JEV

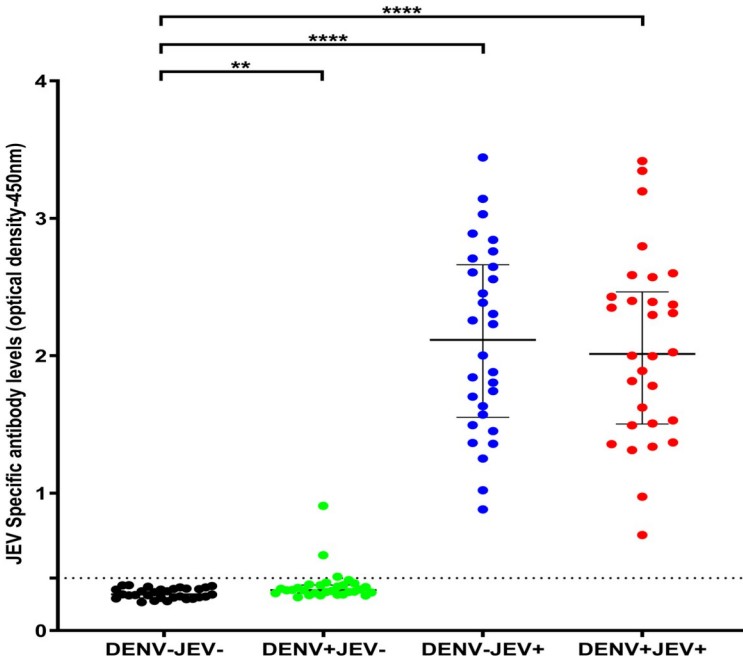

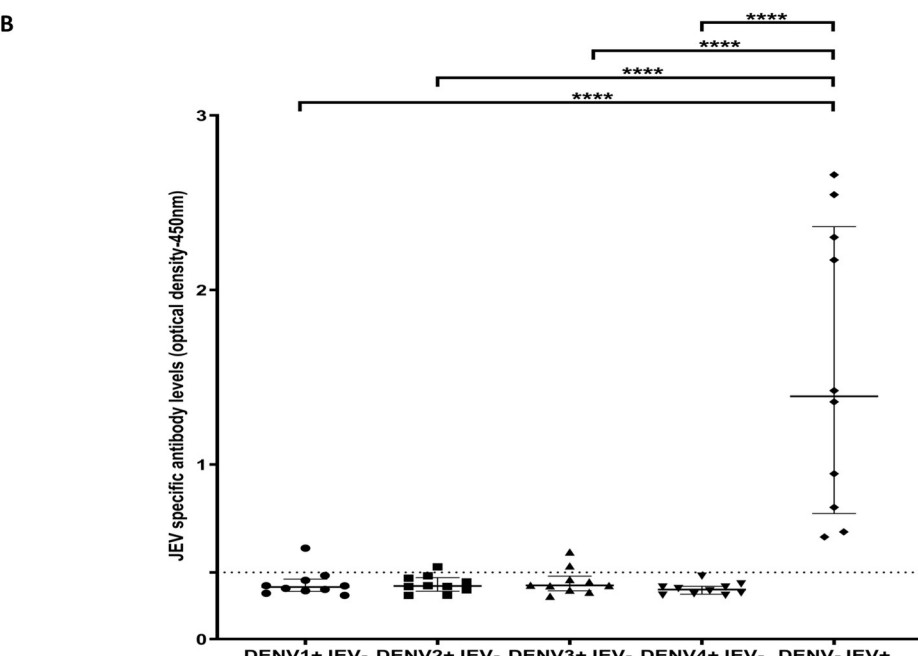

**Fig 2. IgG antibody levels for a JEV-specific peptide pool in individuals with varied DENV and JEV seropositivity.** Antibody responses were measured by ELISA to the JEV pool of peptides (P6, P7, P8, P9, P10, P12 and P13) in (A) those who were both JEV and DENV seropositive (JEV+DEV+, n = 30), JEV seropositive but seronegative for DENV (JEV+DEV-, n = 30), DENV seropositive but JEV seronegative (JEV-DENV+, n = 30) or seronegative for both (JEV-DENV-, n = 30). (B) Who were seronegative for JEV and previously had primary DENV infection due to either DENV1 (n = 10), DENV2 (n = 10), DENV3 (n = 10) or DENV4 (n = 10) and in individuals who were immunized for JEV but DENV seronegative (DENV-JEV+, n = 10) Error bars indicate the median and the interquartile range. The horizontal dotted line represents the cut-off value of 0.382, which was considered as the mean, ±3SD of the optical

density of antibody levels in JEV⁻ and DENV⁻ seronegative individuals. *P<0.05, **P<0.01, ***P<0.001, ****P<0.0001. The differences in the responses to the JEV-specific antibodies were assessed by Mann Whitney two-tailed test.

seronegative but immune to DENV4⁺ individuals responded to this peptide pool. (Fig 2B). The E protein specific responses were measured by the Indirect Panbio IgG ELISA, which has the DENV E protein as the coating antigen and the responses were expressed as PanBio unit. The responses of all 30 individuals is shown S4 Table. There was no difference in the DENV-IgG responses in those 3/30 individuals' who responded to the JEV-specific peptides, compared to those who did not.

## Cross reactivity of these JEV peptides in the convalescent phase of those with acute dengue infection

A much wider repertoire of B cells is known to be stimulated during a secondary dengue infection [26]. There is a higher possibility of generation of cross-reactive antibodies following a secondary dengue infection. Therefore, we proceeded to determine the generation of antibodies that cross react with this JEV-specific pool of 7 peptides following secondary dengue infections. In order to address the possibility of cross reactivity between these JEV-specific peptides and DENV-specific IgG responses, we assessed the responses to JEV peptides in 69 patients, 21–30 days following acute primary and secondary dengue infection. Of these patients 26 (37.7%) had a primary dengue and 43 (62.3%) had secondary dengue infection. 41 (59.4%) had DF and 28 (40.6%) had DHF. In addition, as individuals over 30 years of age are likely to have received the JEV vaccine as they were born after 1989 when the JEV vaccine was introduced to Sri Lanka, we also analyzed responses to these peptide pool based on their age [27]. 38 (55.1%) of the 69 individuals were over 30 years of age and 31 (44.9%) were ≤30 years (and probably immune to the JEV through vaccination) (Table 1).

Of those with a secondary dengue infection, 21 (48.8%) responded to the pool of JEV-specific peptides and 14 (66.7%) were ≤ 30 years of age and 7 (33.3%) were over 30 years. Only 1 individual ≤ 30 years with secondary dengue did not respond to the JEV peptides. Therefore, those who were ≤ 30 years of age were significantly (p<0.0001) more likely to respond to the JEV-specific peptides than those who were older (odds ratio = 42.0, 95% CI 4.88 to 460.8) following an acute secondary dengue infection, presumably due to pre-existing immunity to the JEV.

Similarly, 17 (65.4%) of those with a primary dengue infection responded to the pool of JEV-specific peptides. 2 (11.8%) of these individuals were > 30 years of age and 15 (88.2%) of them were ≤ 30 years of age. Again, those who were ≤ 30 years of age were significantly more likely (p = 0.0002) to respond to the JEV pool of specific peptides compared to those who were >30 years of age (odds ratio = 60.0, 95% CI 4.2 to 688.2). While those who were ≤ 30 years are likely to have had the JEV vaccine, the JE serostatus in those who were above 30 years of age

Table 1. Cross reactivity of these JEV peptides in the convalescent phase of those with acute dengue infection.

| DENV serostatus | Age <30 years N = 31 | Age >30 years N = 38 |
|---|---|---|
| Primary dengue who received JEV vaccine | 15 (48.4%) | 0 (0) |
| Primary dengue who did not receive JEV vaccine | 0 (0) | 2 (5.3%) |
| Secondary dengue who received JEV vaccine | 14 (41.2%) | 0 (0) |
| Secondary dengue who did not receive JEV vaccine | 0 (0) | 7 (18.4%) |

**Table 2. Severity of past severe disease in those who responded to the JEV specific peptides and in those who received the JEV vaccine.**

|  | Past SD (n = 175) | Past NSD (n = 175) | Odds ratio and 95% CI | P value |
|---|---|---|---|---|
| Responded to the JEV-specific peptide pool | 120 (68.6%) | 79 (45.1%) | 5.3, (3.3 to 8.3) | P<0.0001 |
| JEV vaccine received | 84/120 (70.0%) | 50/79 (63.3%) | 2.3, (1.5 to 3.6) | P = 0.0003 |

could not be determined due to the highly cross reactivity of JEV and DENV, especially in acute dengue infection. Those with secondary dengue were not more likely to respond to the pool of JEV-specific peptides compared to those with primary dengue (p = 0.2).

## JEV-specific immune responses in those with varying severity of past infection

Previously, we found that those who had past severe dengue were more likely to be seropositive for JEV [10]. In order to find out if those with past SD were indeed more likely to have JEV-specific IgG antibodies, we used our ELISA with the pool of JEV-specific peptides to assess the JEV-specific antibody responses in 175 individuals who had past SD, and 175 individuals who had past non-severe dengue (NSD) who were age-matched. There was no significant difference (p = 0.12) in the age of those who had past SD (mean = 27.6, SD ± 17.2 years) compared to those with past NSD (mean = 31, SD ± 19.4 years). 120 (68.6%) of those with past SD and 79 (45.1%) of those with past NSD had antibody responded to our JEV-specific pool of peptides (Table 2). Therefore, those who had past SD were significantly more likely (p<0.0001) to have JEV-specific antibodies than those with past NSD (OR 5.3, 95% CI 3.3 to 8.3).

We also assessed the JEV-specific antibody levels to the pool of 7 peptides in those who had past SD compared to those with past NSD and found that individuals with past SD had significantly higher (p<0.0002) antibody level to the JEV-specific pool of peptides (median 0.946, IQR 0.323 to 1.837 OD at 450nm), compared to those with past NSD (median 0.359, IQR 0.283 to 1.479 OD at 450nm) (Fig 3).

## Association between JEV immunization and occurrence of severe dengue

In our previous study we observed that those who had received the JEV vaccine were more likely to be hospitalized when infected with the DENV and only 143/565 (25.3%) of those who received the JEV vaccine were seropositive, whereas 297 (52.6%) were seronegative and 125 (22.1%) showed an equivocal response [10]. Since the majority of children who received the JEV vaccine were seronegative for the JEV using the commercial assay and since occurrence of SD was observed significantly more in those who were seropositive for the JEV, we proceeded to further understand the association of the possible JEV vaccination with occurrence of SD.

In this cohort 84/175 (48.0%) individuals with past SD and 50/175 (28.6%) of those with past NSD had received the JE vaccine. Of those who received the JEV vaccine, 80/84 (95.2%) of those with past SD and 47/50 (94.0%) of those with past NSD had JEV-specific antibody responses to our JEV-specific peptide pool. Therefore, of the total 134 individuals who had received the JEV vaccine, 127 (94.8%) were seropositive for JEV. In contrast, when using the JEV-IgG detection ELISA (Inbios, USA), only 55/84 (67.9%) with past SD and 30/50 (60.0%) with past NSD had detectable JEV-specific antibodies (only 63.4% of those who received the JEV vaccine were seropositive). Therefore, our in-house ELISA using the JEV-specific pool of 7 peptides, appear to be more sensitive than this commercially available ELISA for detection of JEV-specific antibodies following vaccination.

In our cohort, in those who had JEV-specific antibodies, 84/120 (70.0%) with past SD and 50/79 (63.3%) of those with past NSD had received the JEV vaccine. Therefore, this JE

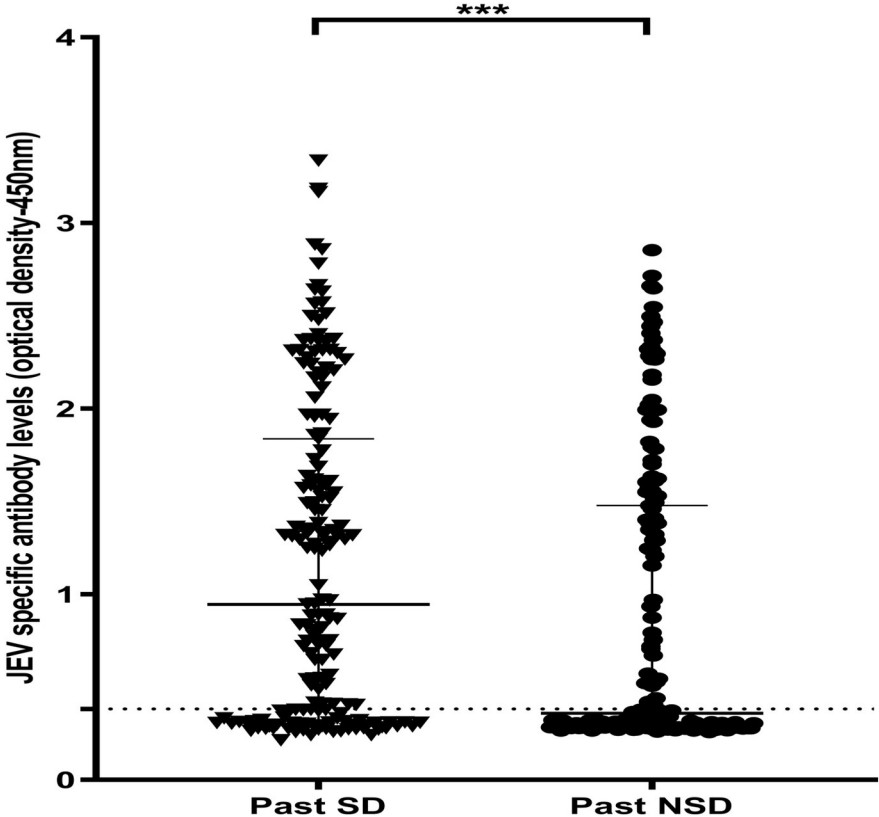

**Fig 3. Antibody levels to JEV-specific pool of peptides in individuals who have had past NSD and SD.** JEV-specific antibody responses were measured by ELISA to the JEV pool of peptides in those who have had NSD (n = 175) and SD (n = 175). Error bars indicate the median and the interquartile range. The horizontal dotted line represents the cut-off value of 0.382, which was considered as the mean, ±3SD of the optical density of antibody levels in JEV⁻ and DENV⁻ seronegative individuals. Differences in the values of IgG antibody responses to JEV-specific peptides in individuals with SD and NSD were compared using the Mann-Whitney U test (two tailed).

vaccination is likely to be the main contributor for the presence of JEV-specific antibodies in both groups (67.3%). Those who had been vaccinated for JEV were more likely to have had past SD compared to those who were not vaccinated (p = 0.0003, OR = 2.3, 95% CI = 1.5 to 3.6) (Table 2).

## Age stratified JEV seroprevalence

Previously, we found that JEV antibody seropositivity significantly and positively correlated with age, which suggested that there could be ongoing exposure to the JEV throughout an individual's life [10]. For instance, previously in the same cohort, we found that at 7 years 12.3% of children were seropositive for JEV, which increased to 42.6% at 16 years. However, the seropositivity for JEV could be due to the presence of cross reactive DENV antibodies, which were giving false positives. In order to find out if the JEV seropositivity truly increased with age, we used our JEV-specific ELISA to determine age-stratified seroprevalence. We found that JEV-specific antibody seropositivity significantly decreased with age (Spearmans' R = -0.85, p = 0.003) (Fig 4). For instance, by 7 years, while 92.3% of children were seropositive for JEV, only 30.2% were seropositive in the 45 to 54-year age group Table 3. Of those ages >30 years of age (who were born before 1987), 64/236 (27.1%) had JEV-specific IgG antibodies, and only 2 of them had received the JEV vaccine (Table 3). In contrast, 212/284 (74.6%) of those <30

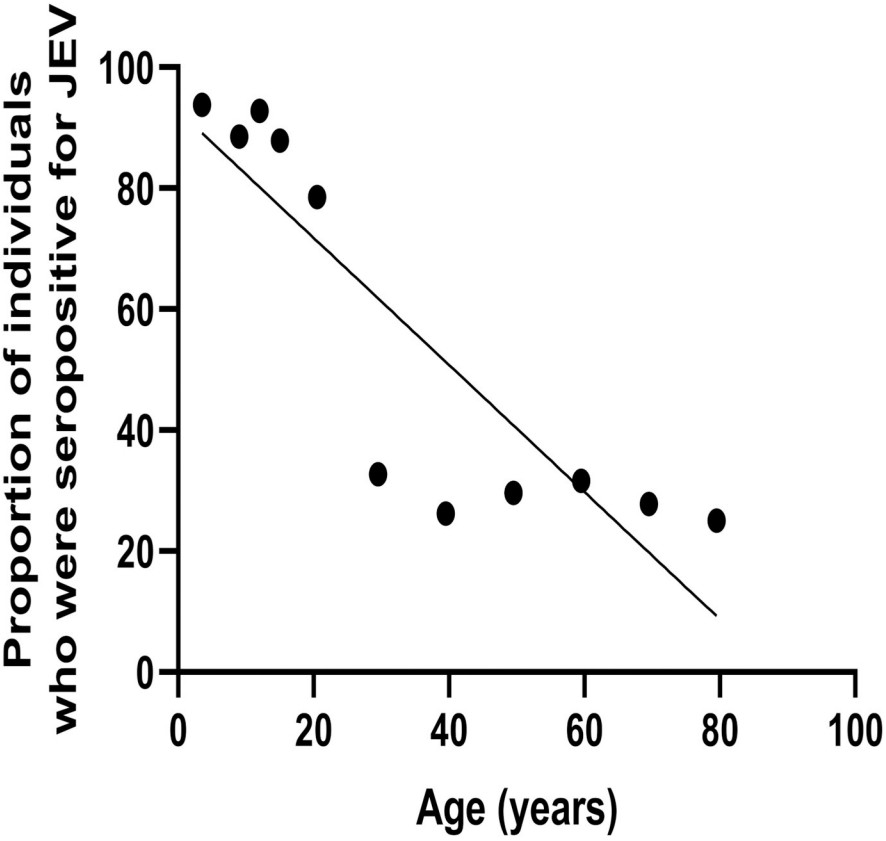

**Fig 4. Age stratified seroprevalence of JE-specific antibodies.** JEV-specific antibodies were measured using the in-house ELISA in 520 healthy individuals. JEV-specific antibody seropositivity significantly decreased with age. The correlation of JEV-specific antibodies with age was measured by the Spearmans' rank correlation (Spearmans' R = -0.85, p = 0.003).

**Table 3. Age stratified detection of anti-JEV antibodies.**

| Age in years | Seropositive N (%) | Seronegative N (%) | Equivocal N (%) | Total |
|---|---|---|---|---|
| <7 | 53 (92.3) | 4 (7.0) | 0 | 57 |
| 8–10 | 19 (86.4) | 3 (13.6) | 0 | 22 |
| 11–13 | 24 (92.4) | 1 (3.8) | 1 (3.8) | 26 |
| 14–16 | 26 (86.7) | 4 (13.7) | 0 | 30 |
| 17–24 | 63 (77.7) | 16 (19.8) | 2 (2.5) | 81 |
| 25–34 | 31 (33.0) | 60 (63.8) | 3 (3.2) | 94 |
| 35–44 | 19 (26.0) | 51 (70.0) | 3 (4.0) | 73 |
| 45–54 | 19 (30.2) | 42 (66.7) | 2 (3.1) | 63 |
| 55–64 | 15 (30.6) | 32 (65.3) | 2 (4.1) | 49 |
| 65–74 | 5 (29.4) | 12 (70.6) | 0 | 17 |
| >75 | 2 (25.0) | 6 (75.0) | 0 | 8 |

We then proceeded to determine the relationship between age stratified JEV seropositivity with that of DENV seropositivity. There was a significant inverse correlation seen between age stratified JEV seroprevalence with that of DENV seroprevalence (p = 0.002, Spearmans R = -0.87).

years of age were seropositive for JEV and of them 202/284 (71.1%) had received the JE vaccine. Therefore, JEV seropositivity was significantly higher in those who were born after 1988 and decreased with advancing age (p = 0.003).

## Discussion

Infection or vaccination with the JEV induce antibody and T cell responses that cross-react with the DENV, thus potentially influencing the disease outcome when infected with the DENV [9, 11]. However, studying how JEV-specific antibody responses influence the outcome of DENV has been difficult due to the cross-reactive nature of the antibody assays that measure JEV antibody responses [13, 14, 28]. In this study, in order to answer this question, we identified JEV-specific peptides from highly conserved regions within the JEV, which do not share homology with the DENV. We found that seven peptides (P6, P7, P8, P9, P10, P12, and P13) representing the envelop region of the JEV were predominantly recognized by those who were immune to JEV (who had received the JEV vaccine). None of the individuals who were seronegative for DENV or JEV responded to this pool consisting of 7 peptides, only one JEV⁻DENV⁺ individual gave positive response for P8 and P13 and none of the JEV⁻DENV⁺ individuals responded to other five peptides (P6, P7, P9, P10, and P12). They had a less than 36% homology with four DENVs (S1 Table and S1 Fig) and all seven peptides, were seen in the exterior surface of the E protein (S2 Fig), with a higher potential to be immunogenic. P6 and P7 were located in the exterior of the JEV E domain III, which is a highly immunogenic domain [29, 30]. It was shown that monoclonal antibodies to this region inhibited infection at a post-attachment step, including blockade of virus fusion and that antibodies with potent neutralizing potential was directed against this region [29–31].

We then proceeded to develop an ELISA using these seven peptides as a peptide pool to differentiate JEV-specific responses from those due to the DENV. While none of the JEV⁻DENV⁻ individuals responded to this peptide pool, all of the JEV⁺DEV⁺ individuals and all of the JEV⁺DENV⁻ individuals responded to this pool of peptides, suggesting that this peptide pool was in fact quite specific to detect JEV-specific antibody responses. Since these peptides appear to be highly immunogenic, and specific, they are likely to be useful to identify JEV-specific immunity in those with pre-existing DENV-immunity.

In order to further evaluate its specificity, we assessed the responses to the pool of JEV peptides in the convalescent period (day 21 since onset of acute infection) in those with both primary and secondary dengue. As a more broadly cross-reactive antibody repertoire has shown to be generated following secondary dengue infection [26], there is a higher possibility that antibodies that cross react with JEV are generated following a secondary dengue infection. We found that those who were ≤ 30 years of age, and therefore who received the JEV vaccine, were significantly more likely to respond to the JEV-specific pool of peptides following both primary 15/17 (88.2%) and secondary dengue infection 14/21 (66.7%), compared to those who were over 30 years of age. This is possibly due to existence of JEV-specific responses in these individuals who were <30 years of age due to JEV vaccination. Of those who were >30 years of age, 7/21 33.3% of those with secondary dengue and 2/17 11.8% of those with primary dengue responded to the JEV-specific peptide pool. As the JEV serostatus/natural exposure to JEV in these individuals prior to the acute dengue infection was not known, it is difficult to interpret these findings whether these responses were due to natural responses or those due to generation of cross-reactive antibodies. However, those with secondary dengue were not more likely to respond to the JEV-specific peptides compared to those with primary dengue, suggesting that the JEV responses following DENV infection was unlikely to be due to the generation of more broadly cross-reactive antibody responses.

Our JEV-specific peptide assay was found to be more sensitive and specific than many commercial assays in detecting JEV-specific antibodies. For instance, in our previous study, 73.4% of those who received the JEV vaccine was found to be seropositive for JEV when using a commercial JEV IgG ELISA [10], whereas, when using our novel-JEV-specific ELISA in the same cohort we found that 94.8% were seropositive. In addition, previously although we found that only 25.3% of children who received the JEV vaccine had seroconverted, these low seroconversion rates are likely to be due to the lower sensitivity of the assay that was used in this study [10]. However, we were only able to compare the antibody responses to the JEV-specific pool of seven peptides to a commercially available JEV IgG ELISA and not the gold standard assay, which is the plaque reduction neutralization assay (PRNT) for detection of JEV-specific immunity. The PRNT for JEV requires use of a live JEV which could not be isolated in recent years in Sri Lanka due to the reduced incidence of JEV infection following widespread vaccination. The live JEV could not be imported to Sri Lanka for this purpose due to certain regulations and therefore, we were unable to compare the responses to the JEV-specific pool of peptides to that of the PRNT.

Earlier, we found that JEV seropositivity significantly and positively correlated with age, whereas with our JEV-specific in-house ELISA (done in the same cohort) we found that actually the JEV seroprevalence decreased with age. While 92.3% of children were seropositive for JEV by 7 years, only 30.2% of those were seropositive by 45 years of age. This is not surprising given that the JEV vaccine was only introduced to Sri Lanka in 1988 [16] and individuals born before 1988, did not receive the vaccine (those who were older than 30 years). Furthermore, although we earlier reported that both DENV and JEV seropositivity increased with age, in this study, we found a significant inverse correlation between DENV and JEV age stratified seropositivity, further suggesting that broadly cross-reactive responses to the DENV do not become positive by our assay. However, measuring JEV antibody levels using cross-reactive antibody assays could lead to the assumption that JEV seropositivity increases with age, possibly due to ongoing exposure. Therefore, this assay would be an important tool in determining JEV seropositivity in populations who are also exposed to the DENV, especially when undertaking dengue vaccine trials.

In this study we found that 45.1% of those with past NSD and 68.6% with past SD had JEV-specific antibody responses, and those who were found to be seropositive for JEV were significantly more likely to have had past SD. Of those who had JEV-specific antibodies, 63.8% had received the JEV vaccine, and therefore, the presence of JEV-specific antibodies in these individuals are more likely to be due to JEV immunization. In Sri Lanka, immunization against the JEV started in a phase by phase basis, and since 1988 was subsequently included in the national routine immunization program [16]. The vaccine is given to all children at approximately one year of age, which preceded the infection with the DENV resulting in DHF in those children. In addition, the JEV seropositivity of older age groups was significantly less, further suggesting that the JEV vaccine was the main contributor the JEV seropositivity in this population. The JEV has been shown to induce ADE at sub-neutralizing concentrations, in vitro [9]. Although pre-existent JEV antibodies due to JEV vaccination has shown to potentiate DENV infection in vitro and although our data suggest that those who had received the JEV vaccine (and therefore had JEV-specific IgG antibodies) were more likely to have severe dengue, these data should be further validated by carrying out careful prospective studies in the communities where these viruses co-circulate. Given that the JEV vaccine has significantly reduced mortality and morbidity of encephalitis due to JEV throughout the world including Sri Lanka [27, 32, 33], it would be important to carefully examine such associations in other JEV and DENV endemic countries before concluding that JEV increases dengue disease severity.

In summary, we have identified JEV-specific peptides from highly conserved regions of the JEV, which do not cross react with DENV-specific IgG antibody responses. Using these peptides, we have developed a novel in-house ELISA to detect JEV-specific IgG antibodies, which showed that those with pre-existing antibodies to the JEV were more likely to develop severe clinical disease when infected with the DENV. Since there is a possibility that JEV-specific antibodies may modulate the immune response to the DENV, large prospective cohort studies should be carried out to further understand such associations.

## Supporting information

**S1 Fig. Seven peptides homology with four DENVs.**
(TIF)

**S2 Fig. Seven peptides on JEV E protein.**
(TIF)

**S1 Table. The homology of JEV specific commercially synthesized peptides with four dengue serotypes, WNV, YFV and Zika virus.**
(DOCX)

**S2 Table. JEV specific 22 peptides antibody responses for individuals who have had varied JEV and DENV seropositivity.**
(DOCX)

**S3 Table. Epitope prediction by four B cell linear epitope prediction servers (Bepipred, Elipro, BCEpred, and ABCpred).**
(DOCX)

**S4 Table. PanBio units of JEV⁻DENV⁺ thirty individuals.** (DOCX)

$$\text{S4 Table. PanBio units of } JEV^{-}DENV^{+} \text{ thirty individuals.}$$
(DOCX)

**S1 File. Classification of past dengue disease severity in our cohort of healthy individuals and a novel T cell-based assay to detect past infecting DENV serotype.**
(DOCX)

## Author Contributions

**Conceptualization:** Gathsaurie Neelika Malavige, Charitha Goonesekara.

**Data curation:** Pradeep Darshana Pushpakumara, Chandima Jeewandara, Yashodha Perera, Ananda Wijewickrama.

**Formal analysis:** Pradeep Darshana Pushpakumara, Gathsaurie Neelika Malavige, Charitha Goonesekara.

**Funding acquisition:** Gathsaurie Neelika Malavige, Charitha Goonesekara.

**Investigation:** Pradeep Darshana Pushpakumara, Chandima Jeewandara, Laksiri Gomes, Yashodha Perera.

**Methodology:** Chandima Jeewandara, Laksiri Gomes, Ananda Wijewickrama.

**Project administration:** Gathsaurie Neelika Malavige, Charitha Goonesekara.

**Resources:** Ananda Wijewickrama, Gathsaurie Neelika Malavige.

**Supervision:** Gathsaurie Neelika Malavige, Charitha Goonesekara.

**Writing – original draft:** Pradeep Darshana Pushpakumara, Gathsaurie Neelika Malavige, Charitha Goonesekara.

**Writing – review & editing:** Gathsaurie Neelika Malavige, Charitha Goonesekara.

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
