## [Decision Letter · Decision Letter 0]

10 Sep 2020

PONE-D-20-25640

Development and validation of an assay for detection of Japanese Encephalitis virus specific antibody responses

PLOS ONE

Dear Dr. Malavige,

Thank you for submitting your manuscript to PLOS ONE. After careful consideration, we feel that it has merit but does not fully meet PLOS ONE’s publication criteria as it currently stands. Therefore, we invite you to submit a revised version of the manuscript that addresses the points raised during the review process.

While both reviewers indicate their enthusiasm for the manuscript, they each of some concerns with regards to the use of linear peptides and its limitations and the potential use of available proteins that could be used for defining conformational epitopes. Please respond to each of the issues raised in your revised manuscript.

We look forward to receiving your revised manuscript.

Kind regards,

Aftab A. Ansari, PhD

Academic Editor

PLOS ONE

Journal Requirements:

"Funding was provided by the Centre for Dengue Research and National Science Foundation, Sri Lanka (RG/2015/HS/07)."

3. We note you have included a table to which you do not refer in the text of your manuscript. Please ensure that you refer to Table 3 in your text; if accepted, production will need this reference to link the reader to the Table.

Reviewers' comments:

Reviewer's Responses to Questions

**Comments to the Author**

1. Is the manuscript technically sound, and do the data support the conclusions?

Reviewer #1: Yes

Reviewer #2: Partly

2. Has the statistical analysis been performed appropriately and rigorously? 

Reviewer #1: Yes

Reviewer #2: No

3. Have the authors made all data underlying the findings in their manuscript fully available?

Reviewer #1: Yes

Reviewer #2: Yes

4. Is the manuscript presented in an intelligible fashion and written in standard English?

Reviewer #1: Yes

Reviewer #2: Yes

5. Review Comments to the Author

Reviewer #1: In this manuscript,the authors developed an ELISA that can effeciently distinguish JEV infection from the closely related DENV. The authors identified 20 highly conserved peptides within the JEV and screen patient data with various flavivirus exposures. The authors identified a JEV-specific peptide pools that did not cross-react with DENV. They then used this ELISA to screen subjects with prior DENV exposure to determine the level of JEV immunity in these individuals. This is an excellent and comprehensive study that provides important information regarding JEV expsoure history (through vaccination or otherwise) in dengue-infected patients and highlights the potential of cros-reactive responses that could impact antibody responses.

Comments

1. While the seropositivity information is provided for the dengue subjects, what are the actual E protein titers for the individual strains in these subjects? How does this compare with the level of cross-reactivity with the JEV peptides in the pool?

Reviewer #2: Pushpakumara et al examine JEV immune responses from individuals who have been exposed to JEV and/or DENV. This study utilizes a broad panel of JEV peptides to determine immunodominant peptide epitopes. From this data, the study attempts to correlate certain JEV immune responses to DHF. The ability to rapidly determine JEV seroconversion is important and the development of such assays such as in this study is very important. However, this manuscript would be much more effective if the authors take into consideration and compare their results to prior JEV antibody studies as well as examine immune responses not only towards linear peptide epitopes but to conformational epitopes as well.

Major comments

1. Given the number of peptides used in this study and the JEV/DENV comparison studies, it would be extremely helpful to include where these peptides map to on the E protein. How much homology exists for these peptides between JEV and DENV? Existing structural data (PDB 3P54) can be used to map these peptide epitopes on the JEV E protein structure.

2. A highly conserved peptide epitope between JEV and DENV (and other flaviviruses) is the fusion loop. Were antibody responses observed against the fusion loop? And if so, how many of the individuals exhibited such responses?

3. Previous studies have examined antibody responses against JEV and have identified protective and/or immunogenic epitopes (PMID 12551998, 29487230, 18480437, 2432163). One of the identified protective epitopes is domain III on the envelope protein. The authors include several peptides that map to domain III but it would be quite informative if the authors discussed how their DIII peptide epitopes relate to those epitopes that have previously identified. Are there any overlapping immunodominant epitopes?

4. The authors state that a limitation to their study is the inability to use live JEV virus to characterize immune responses and thereby only use peptide-based approach. This is a limitation as it only examines immune responses against linear epitopes, and not against conformational epitopes. Several JEV protective antibodies have been identified to conformational epitopes (e.g. DIII lateral ridge, E dimer interface). An alternative method to characterize JEV-specific immune responses, although not neutralization based, would be to look at immune responses against a more conformationally relevant antigen such as the recombinant JEV E protein (https://thenativeantigencompany.com/products/japanese-encephalitis-virus-envelope-protein/).

5. The authors mention that individuals who have been vaccinated for JEV have more severe Dengue disease yet in these individuals, the antibody responses are against JEV-specific peptides that do not share homology with DENV (What % homology is shared—if any). The authors additionally claim that JEV-specific antibodies may modulate the immune response to DENV. Could the authors further discuss how JEV-specific antibodies—which do not cross-react with DENV—modulate DENV pathogenesis?

6. The authors did not include which statistical tests were done for the experiments. This is critical and must be included in the figure legends.

6. PLOS authors have the option to publish the peer review history of their article (what does this mean?). If published, this will include your full peer review and any attached files.

Reviewer #1: No

Reviewer #2: No

---

## [Author Response · Author response to Decision Letter 0]

21 Sep 2020

21th September 2020

Dr. Aftab A. Ansari,

Academic Editor,

PLOS ONE.

Dear Dr. Ansari,

Submission of a manuscript titled ‘Development and validation of an assay for detection of Japanese Encephalitis virus specific antibody responses’ 

We wish to thank the reviewers for carefully going through our manuscript and the useful suggestions they have made. We have incorporated all these changes in the revised version and letter and have addressed all issues raised by the reviewers. We have highlighted the changes in the revised manuscript.

Reviewer 1: 

Reviewer #1: In this manuscript, the authors developed an ELISA that can efficiently distinguish JEV infection from the closely related DENV. The authors identified 20 highly conserved peptides within the JEV and screen patient data with various flavivirus exposures. The authors identified a JEV-specific peptide pools that did not cross-react with DENV. They then used this ELISA to screen subjects with prior DENV exposure to determine the level of JEV immunity in these individuals. This is an excellent and comprehensive study that provides important information regarding JEV exposure history (through vaccination or otherwise) in dengue-infected patients and highlights the potential of cros-reactive responses that could impact antibody responses.

Comments

1. While the seropositivity information is provided for the dengue subjects, what are the actual E protein titers for the individual strains in these subjects? How does this compare with the level of cross-reactivity with the JEV peptides in the pool?

Response: We thank the reviewer for the comment. All the JEV specific peptides of the pool belong to the E protein of the JEV. Sixty out 60 JEV+ individuals pre-existing Abs against JEV successfully identified by this pool of seven peptides. Furthermore, 3/30 JEV-DENV+ individuals responded to the seven-peptide pool. We did not measure the E protein measures by an in-house ELISA but used the PanBio Indirect IgG ELISA as a surrogate marker of antibody responses to the E protein, as the coating antigen in that kit is the E protein. There was no significant difference in the PanBio (IgG) units of these 3 JEV-DENV+ individuals who responses to the JEV specific peptides, compared to others (S4 Table). All the individual values of these 30 individuals are indicated in supplementary table 4 (S4 table). 

Reviewer #2: Pushpakumara et al examine JEV immune responses from individuals who have been exposed to JEV and/or DENV. This study utilizes a broad panel of JEV peptides to determine immunodominant peptide epitopes. From this data, the study attempts to correlate certain JEV immune responses to DHF. The ability to rapidly determine JEV seroconversion is important and the development of such assays such as in this study is very important. However, this manuscript would be much more effective if the authors take into consideration and compare their results to prior JEV antibody studies as well as examine immune responses not only towards linear peptide epitopes but to conformational epitopes as well. 

Response

We thank the reviewer for these very valuable comments. It would be extremely valuable to examine the immune responses to conformational epitopes as well. However, this was beyond the scope of this study. The patients in whom the JEV specific immune responses were assessed, we investigated using commercially available JEV ELISA as well (Jeewandara et al, PLOS One, 2015). These samples were collected in 2013 and we have compared the JEV specific antibody responses to the commercially available JEV ELISA, and the JEV-specific peptide pools identified by us. We found that the JEV-specific pool of peptides identified by us, was more sensitive and far more specific that the previously used JEV commercial ELISA. We understand that by only using linear peptides, we are likely to miss potentially important conformational epitopes. 

Major comments

1. Given the number of peptides used in this study and the JEV/DENV comparison studies, it would be extremely helpful to include where these peptides map to on the E protein. How much homology exists for these peptides between JEV and DENV? Existing structural data (PDB 3P54) can be used to map these peptide epitopes on the JEV E protein structure.

We thank the reviewer for this important question. In this study we screened antibody responses to 22 JEV peptides in individuals who have had varied JEV and DENV seropositivity. We have indicated in supplementary figure 2 (S2 Fig) of the revised version of the manuscript where the 7 peptides map on the E protein. The homology between these 7 peptides and the DENV is shown in S1 table of the revised version of the manuscript. 

All these seven peptides show less than 40% homology with four DENVs (S1 Table and S1 Fig). These peptides shown higher specificity for JEV during the ELISpot assays. According to the previous studies, linear JEV peptides which have more than 60% homology with WNV viruses also gave JEV specific antibody responses (Masrinoul et al., 2011)(R.-H. Hua et al., 2013)(R. H. Hua et al., 2010). 

2. A highly conserved peptide epitope between JEV and DENV (and other flaviviruses) is the fusion loop. Were antibody responses observed against the fusion loop? And if so, how many of the individuals exhibited such responses?

Response: Thank you again for this important question. Fusion loop is located on between 98-110 residues (Allison et al., 2001), and none of above seven peptides are located on fusion loop. Therefore, antibody responses against the fusion loop was not observed. We have included this information in the revised version of the manuscript. 

3. Previous studies have examined antibody responses against JEV and have identified protective and/or immunogenic epitopes (PMID 12551998, 29487230, 18480437, 2432163). One of the identified protective epitopes is domain III on the envelope protein. The authors include several peptides that map to domain III but it would be quite informative if the authors discussed how their DIII peptide epitopes relate to those epitopes that have previously identified. Are there any overlapping immunodominant epitopes?

JEV domain III is a highly immunogenic region in envelope protein (Lin & Wu, 2003)(Fernandez et al., 2018)(Goncalvez et al., 2008) and (Allison et al, 2001). However, only 2/7 peptides (P6 and P7) belong to domain III. The immunogenicity of these two regions represented by these peptides have previously been identified by using neutralizing monoclonal antibodies (MAbs) (Fernandez et al., 2018). Many of these MAbs inhibited infection at a post-attachment step, including blockade of virus fusion. Mapping studies using site-directed mutagenesis and hydrogen-deuterium exchange with mass spectrometry revealed that the lateral ridge on domain III of the envelope protein was a primary recognition epitope for our panel of strongly neutralizing MAbs (Fernandez et al., 2018). 

Allison, S. L., Schalich, J., Stiasny, K., Mandl, C. W., & Heinz, F. X. (2001). Mutational Evidence for an Internal Fusion Peptide in Flavivirus Envelope Protein E. Journal of Virology, 75(9), 4268–4275. https://doi.org/10.1128/JVI.75.9.4268-4275.2001

Fernandez, E., Kose, N., Edeling, M. A., Adhikari, J., Sapparapu, G., Lazarte, S. M., Nelson, C. A., Govero, J., Gross, M. L., Fremont, D. H., Crowe, J. E., & Diamond, M. S. (2018). Mouse and human monoclonal antibodies protect against infection by multiple genotypes of japanese encephalitis virus. MBio, 9(1). https://doi.org/10.1128/mBio.00008-18

Goncalvez, A. P., Chien, C.-H., Tubthong, K., Gorshkova, I., Roll, C., Donau, O., Schuck, P., Yoksan, S., Wang, S.-D., Purcell, R. H., & Lai, C.-J. (2008). Humanized Monoclonal Antibodies Derived from Chimpanzee Fabs Protect against Japanese Encephalitis Virus In Vitro and In Vivo. Journal of Virology, 82(14), 7009–7021. https://doi.org/10.1128/jvi.00291-08

Hua, R.-H., Liu, L.-K., Chen, Z.-S., Li, Y.-N., & Bu, Z.-G. (2013). Comprehensive Mapping Antigenic Epitopes of NS1 Protein of Japanese Encephalitis Virus with Monoclonal Antibodies. PLoS ONE, 8(6), e67553. https://doi.org/10.1371/journal.pone.0067553

Hua, R. H., Chen, N. S., Qin, C. F., Deng, Y. Q., Ge, J. Y., Wang, X. J., Qiao, Z. J., Chen, W. Y., Wen, Z. Y., Liu, W. X., Hu, S., & Bu, Z. G. (2010). Identification and characterization of a virus-specific continuous B-cell epitope on the PrM/M protein of Japanese encephalitis Virus: Potential application in the detection of antibodies to distinguish Japanese encephalitis Virus infection from West Nile. Virology Journal, 7, 1–8. https://doi.org/10.1186/1743-422X-7-249

Lin, C. W., & Wu, S. C. (2003). A functional epitope determinant on domain III of the Japanese encephalitis virus envelope protein interacted with neutralizing-antibody combining sites. J Virol, 77(4), 2600–2606. https://doi.org/10.1128/JVI.77.4.2600-2606.2003

Masrinoul, P., Diata, M. O., Pambudi, S., Limkittikul, K., Ikuta, K., & Kurosu, T. (2011). Highly conserved region 141-168 of the NS1 protein is a new common epitope region of Dengue virus. Japanese Journal of Infectious Diseases, 64(2), 109–115.

4. The authors state that a limitation to their study is the inability to use live JEV virus to characterize immune responses and thereby only use peptide-based approach. This is a limitation as it only examines immune responses against linear epitopes, and not against conformational epitopes. Several JEV protective antibodies have been identified to conformational epitopes (e.g. DIII lateral ridge, E dimer interface). An alternative method to characterize JEV-specific immune responses, although not neutralization based, would be to look at immune responses against a more conformationally relevant antigen such as the recombinant JEV E protein (https://thenativeantigencompany.com/products/japanese-encephalitis-virus-envelope-protein/).

Response: We thank the reviewer for this suggestion. Using a recombinant protein will indeed help us to map conformational epitopes. However, it would not help us to map JEV specific epitopes that do not cross react with the DENV, as the whole E protein has almost 50% homology with four DENVs, Therefore, in this study we initially carried out bioinformatics analysis to determine JEV specific peptides that do not cross react with the four DENVs. According to the linear B cell epitope prediction tools, these 7 peptides antigenicity were successfully determined (S3 Table). 

5. The authors mention that individuals who have been vaccinated for JEV have more severe Dengue disease yet in these individuals, the antibody responses are against JEV-specific peptides that do not share homology with DENV (What % homology is shared—if any). The authors additionally claim that JEV-specific antibodies may modulate the immune response to DENV. Could the authors further discuss how JEV-specific antibodies—which do not cross-react with DENV—modulate DENV pathogenesis?

Response

Response: We apologize for the lack of clarity. This pool consisting of 7 JEV specific peptides were used to identify ‘specific’ JEV specific immunity. Our results showed that those who responses to this peptide pool consisting of these 7 peptides (and therefore, likely to be infected or vaccinated with the JEV), were more likely to have severe dengue. The peptide pool was used to identify those who were immune to JEV (by infection or by vaccination) vs cross reactive immunity to the DENV. We did not assess the immune responses to these peptides (which were largely JEV-specific), in causing severe dengue. 

6. The authors did not include which statistical tests were done for the experiments. This is critical and must be included in the figure legends.

Response

We apologies for this omission. We have included this in the revised version of the manuscript.

Thank you for considering our manuscript with Journal of Biomedical Science. 

Yours Sincerely,

Prof. Neelika Malavige

---

## [Editor Report · Decision Letter 1]

28 Sep 2020

Development and validation of an assay for detection of Japanese Encephalitis virus specific antibody responses

PONE-D-20-25640R1

Dear Dr. Malavige,

We’re pleased to inform you that your manuscript has been judged scientifically suitable for publication and will be formally accepted for publication once it meets all outstanding technical requirements.

Kind regards,

Aftab A. Ansari, PhD

Academic Editor

PLOS ONE
---

## [Editor Report · Acceptance letter]

6 Oct 2020

PONE-D-20-25640R1 

Development and validation of an assay for detection of Japanese Encephalitis virus specific antibody responses 

Dear Dr. Malavige:

I'm pleased to inform you that your manuscript has been deemed suitable for publication in PLOS ONE. Congratulations! Your manuscript is now with our production department. 

Kind regards, 

on behalf of

Dr. Aftab A. Ansari 

Academic Editor

PLOS ONE